# A boron-transfer mechanism mediating the thermally induced revival of frustrated carbene–borane pairs from their shelf-stable adducts

Yoichi Hoshimoto [1✉], Mahiro Sakuraba[1], Takuya Kinoshita[1], Masaki Ohbo[2], Manussada Ratanasak[2], Jun-ya Hasegawa[2✉] & Sensuke Ogoshi [1✉]

Chemists have designed strategies that trigger the conformational isomerization of molecules in response to external stimuli, which can be further applied to regulate the complexation between Lewis acids and bases. We have recently developed a system in which frustrated carbene–borane pairs are revived from shelf-stable but external-stimuli-responsive carbene–borane adducts comprised of N-phosphine-oxide-substituted imidazolylidenes (PoxIms) and triarylboranes. Herein, we report the detailed mechanism on this revival process. A thermally induced borane-transfer process from the carbene carbon atom to the N-phosphinoyl oxygen atom initiates the transformation of the carbene–borane adduct. Subsequent conformational isomerization via the rotation of the N-phosphinoyl group in PoxIm moieties eventually leads to the revival of frustrated carbene–borane pairs that can cleave H$_2$. We believe that this work illustrates an essential role of dynamic conformational isomerization in the regulation of the reactivity of external-stimuli-responsive Lewis acid-base adducts that contain multifunctional substituents.

[1] Department of Applied Chemistry, Faculty of Engineering, Osaka University, Osaka, Japan. [2] Institute for Catalysis, Hokkaido University, Sapporo, Japan.
✉email: hoshimoto@chem.eng.osaka-u.ac.jp; hasegawa@cat.hokudai.ac.jp; ogoshi@chem.eng.osaka-u.ac.jp

There have been many recent developments in the chemistry of frustrated Lewis pairs (FLPs) that have been of note, for example, the activation of $H_2$ mediated by main-group elements[1–9]. In general, FLPs are transient and not shelf-stable species making their isolation challenging. Meanwhile, chemists have developed strategies that trigger the conformational isomerization of molecules in response to external-stimuli[10–13]. These strategies can also be used to generate transient FLP species from classical Lewis adducts (CLAs) that act like their shelf-stable precursors[14–26]. In 2015, we demonstrated a strategy to generate FLPs from shelf-stable CLAs (**PoxIm·B[1]** in Fig. 1) that are comprised of *N*-phosphine-oxide-substituted imidazolylidenes (PoxIms; **1**) and B(C₆F₅)₃ (**B[1]**). Here, the revival of the FLP from the CLA is closely controlled by a thermally induced conformational isomerization of the *N*-phosphinoyl moiety[17,27–30]. In 2018, Stephan et al. reported a system to control the generation of FLPs from CLAs via a light-induced *E/Z* isomerization of $(C_6F_5)_2B((p\text{-Tol})S)C = CCH(^tBu)$[31]. Nevertheless, such FLP revival systems, including external-stimuli-responsive conformational isomerizations, are still underdeveloped. Thus, clarifying the relationship between external-stimuli-responsive conformational isomerizations and the interconversion that occurs between frustrated and quenched Lewis pairs is of great importance. This would allow a significant expansion of different strategies to design and apply FLP species[19].

In our system that uses PoxIms, the revival mechanism has not been fully explained. A tentative mechanism in which a $B(C_6F_5)_3$ moiety is repelled by the *N*-phosphinoyl group via a thermally induced isomerization from the *syn* to *anti* conformation had been proposed. In this case, the *syn/anti* conformation refers to the relative orientation of the carbene carbon atom and the *N*-phosphinoyl oxygen atom with respect to the N–P bond (Fig. 1a)[17]. Herein, we report the results of a combined experimental and theoretical mechanistic study that demonstrates the key role of a transfer step where the triarylborane (BAr₃) unit on the carbene carbon atom moves to the *N*-phosphinoyl oxygen atom (Fig. 1b). In this study, PoxIms with $2,6\text{-}^i Pr_2\text{-}C_6H_3$, $2,4,6\text{-}Me_3\text{-}C_6H_2$, and $3,5\text{-}^t Bu_2\text{-}C_6H_3$ groups were studied and are herein referred to as **1a**, **1b**, and **1c**, respectively.

## Results and discussion

**Effects of Lewis acidity**. To explore the impact of the Lewis acidity of BAr₃ on the formation and reactivity of the carbene–borane adducts, the reaction between **1a** and B(*p*-HC₆F₄)₃ (**B[2]**) was undertaken (Fig. 2a). Full consumption of **1a** was confirmed after 20 min, resulting in the formation of two CLAs, i.e., **2aB[2]**, which contains a *N*-phosphinoyl oxygen–boron bond, and **3aB[2]**, which contains a carbene–boron bond, in 61% and 29% yield, respectively. Previously, we have reported that, even at −30 °C, **2aB[1]** could be converted to **3aB[1]** and that full identification of **2aB[1]** could therefore be achieved using NMR analysis conducted at −90 °C[17]. In the present case, **2aB[2]** exhibited a longer life-time at room temperature than **2aB[1]**, which enabled us to prepare single crystals of **2aB[2]** by recrystallization from the reaction mixture at −30 °C. The molecular structure of **2aB[2]** was unambiguously confirmed using single-crystal X-ray diffraction (SC-XRD) analysis. A set of ($R_a$) and ($S_a$) atropisomers of **2aB[2]** was identified in the asymmetric unit of the single crystal. The molecular structure of ($R_a$)-**2aB[2]** is shown in Fig. 2b and demonstrates a rare example of complexation-induced N–P axial chirality[29]. As the reaction progressed, **2aB[2]** was converted to **3aB[2]** and **4a**; **2aB[2]** was fully consumed within 6 h to afford these compounds in 75% and 25% yield, respectively. It should be noted that **4a** is likely furnished via the migration of the *N*-phosphinoyl group from the nitrogen atom to the carbene carbon atom. However, in the absence of **B[2]**, this migration only proceeded to 9% at 100 °C, even after 25 h[30]. The formation of **4a** was therefore promoted by the enhancement of the electrophilicity of the P center via the coordination of the *N*-phosphinoyl moiety to **B[2]**. Regeneration of **B[2]** was observed along with the production of **4a**. The molecular structure of **3aB[2]** was also confirmed by SC-XRD analysis (Fig. 2c). Comparison of the structural parameters between the solid-state structures of **3aB[2]** and **3aB[1]** shows their similarity. For example, the C1–B distances in **3aB[2]** and in **3aB[1]** are 1.710(3) Å and 1.696(3) Å, respectively. The interatomic distance of 3.257(3) Å between the O and B atoms in **3aB[2]** suggests the absence of a specific interaction between these atoms, similar to that in **3aB[1]** (3.234(3) Å).

Thermolysis of **3aB[2]** at 60 °C for 3 h resulted in the generation of **4a** and **B[2]** in 77% and 73% yield, respectively, with concomitant formation of [**1a**-H][HO(**B[2]**)₂] in 4% yield (conversion of **3aB[2]** = 81%; Fig. 3a). Although **2aB[2]** was not observed via NMR analysis of this reaction at 60 °C, the formation of **4a** and **B[2]** indicates the in situ regeneration of **2aB[2]** (*vide supra*). The formation of [**1a**-H][HO(**B[2]**)₂] can be rationalized in terms of a reaction between contaminated $H_2O$ and the FLP species regenerated from **3aB[2]** via **2aB[2]**. The regeneration of the FLP species from **3aB[2]** was then clearly confirmed by treating **3aB[2]** with $H_2$ (5 atm) at 22 °C, resulting in the formation of [**1a**-H][H-**B[2]**] (**5aB[2]**) in 19% yield with concomitant formation of [**1a**-H][HO(**B[2]**)₂] (8%) and **1a** (6%) (Fig. 3b). Under identical conditions, no reaction occurred when **3aB[1]** was used[17]. At 60 °C, **5aB[2]** was generated in 90% yield after 3 h, which is almost comparable with the production of **5aB[1]** (89%) from **3aB[1]**. Thus, the lower Lewis acidity of **B[2]** relative to **B[1]** allowed a more facile revival of the FLP species from **3aB[2]** than from **3aB[1]**. However, the lower Lewis acidity did not affect the progress of the heterolytic cleavage of $H_2$ by FLPs at 60 °C.

**Kinetic studies**. To gain further insight into the reaction mechanism, the initial rate constants for the generation of **5aB[1]**, $k_{int}$ [$10^{-5}$ s⁻¹], from the reaction between **3aB[1]** and $H_2$ in 1,2-dichloroethane-$d_4$ (DCE-$d_4$) at 60 °C were estimated by varying the $H_2$ pressure from 0.5 to 5.0 atm (Fig. 4a). It should be noted here that when $H_2$ was pressurized at 5.0 atm, an excess of $H_2$ (ca.

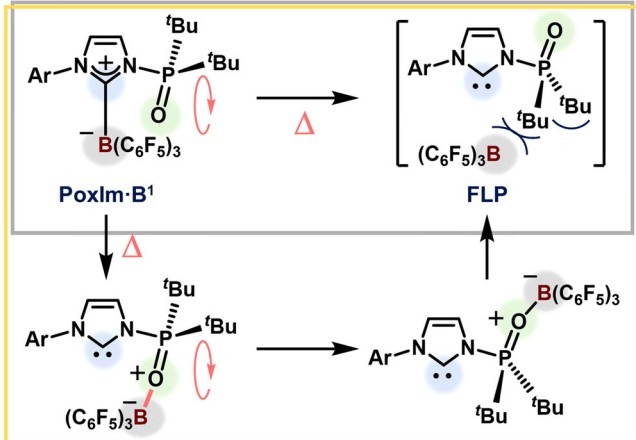

**a** *Originally proposed "rotation-repulsion" mechanism*

**PoxIm·B[1]**          **FLP**

**b** *Updated "jump-rotation" mechanism (this work)*

**Fig. 1 Revival of FLPs from PoxIm·B[1] adducts, induced by thermally responsive molecular motions. a** A previously proposed mechanism. **b** The updated mechanism proposed based on the results of this work.

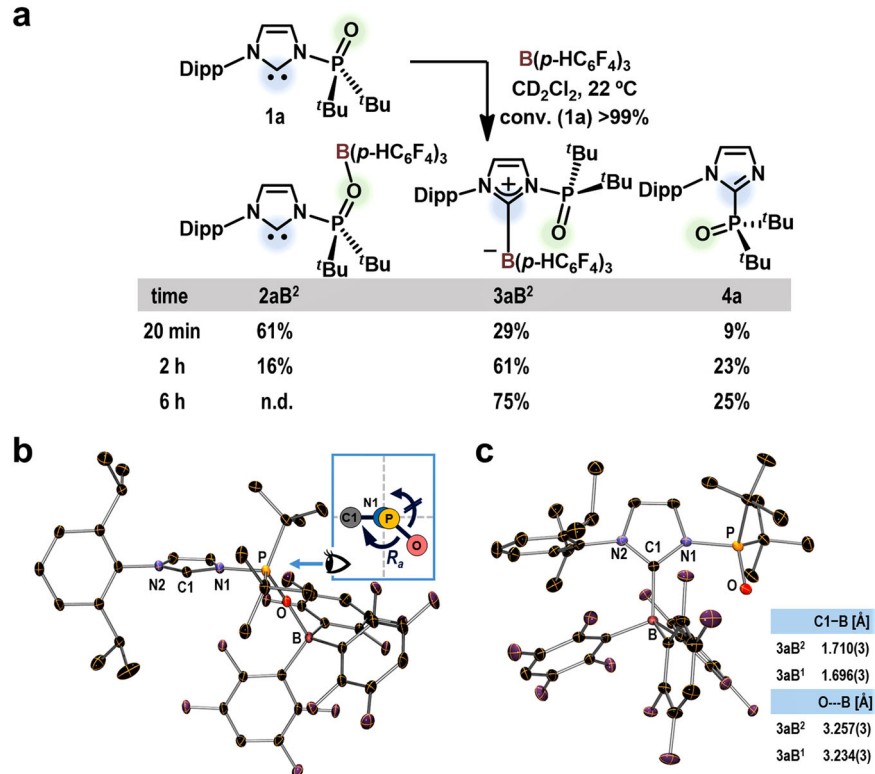

**Fig. 2 Reaction between 1a and B(p-HC6F4)3 (B2). a** The reaction was monitored by NMR spectroscopy and the product yields were estimated based on $^{31}$P NMR analyses. **b** Molecular structure of ($R_a$)-**2aB2** with thermal ellipsoids at 30% probability; H atoms and solvated $C_7H_8$ molecules are omitted for clarity. Selected bond lengths [Å] and angles [°]: O–B 1.556(2), N1–P 1.707(2), P–O 1.513(1); P-O-B 165.2(1), C1-N1-P-O 128.0(1). **c** Molecular structure of **3aB2** with thermal ellipsoids at 30% probability; H atoms are omitted for clarity. For comparison with **3aB1** (cf. ref. [17]), the carbene–boron bond lengths and interatomic distances between oxygen and boron atoms are shown; C1-N1-P-O: 15.3(2)°.

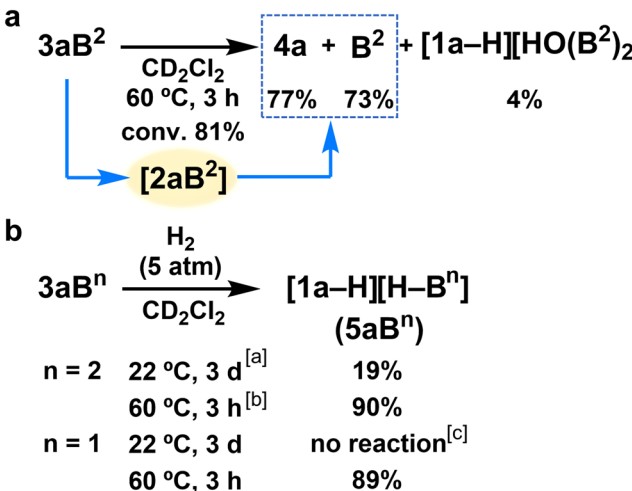

**Fig. 3 Reactivity of carbene–borane adducts 3aB^n (n = 1, 2). a** Thermolysis of **3aB2** monitored via NMR spectroscopy. Product yields were calculated based on $^{31}$P and $^{19}$F NMR analyses. **b** Reaction between **3aB^n** and $H_2$. Product yields were calculated based on $^{19}$F and $^{31}$P analysis. [a][**1a–H**][HO(**B2**)2] and **1a** were also observed in 8% and 6% yield, respectively. [b][**1a–H**][HO(**B2**)2] was also observed in 7% yield. [c]Results obtained using **3aB1** are reproduced from ref. [17].

0.3 mmol) with respect to **3aB1** (0.010 mmol) was added to the pressure-tight NMR tube. The concentration of $H_2$ clearly influenced the progress of the reaction, suggesting that the heterolytic cleavage of $H_2$ by the FLP species is involved in the rate-determining events. Next, the reaction between **3aB1** and $H_2$ at

5.0 atm of pressure was monitored in DCE-$d_4$ whilst the temperature was varied from 50 to 80 °C (Supplementary Figure 27). Pseudo-first order rate constants, $k_{obs}$ [$10^{-5}$ s$^{-1}$], of 2.95(2), 11.2(8), 46.4(4) and 183(2) were estimated for the reactions at 50, 60, 70, and 80 °C, respectively. Thus, the activation energy and pre-exponential factor obtained from the plot based on the Arrhenius equation, $\ln k_{obs} = -(E_a/R)(1/T) + \ln A$, are $E_a = 31.2$ [kcal mol$^{-1}$] and $A = 3.3(36) \times 10^{16}$ [s$^{-1}$] (Fig. 4b). Given the close relation between $E_a$ and $\Delta H^{\ddagger}$, the values obtained for $E_a$ suggest that the formation of **5aB1** via the reaction between **3aB1** and $H_2$ only occurs at temperatures higher than 25 °C[32].

Based on the results presented here and those previously reported[17], the reaction between the carbene–borane adducts and $H_2$ to give [PoxIm-H][H-BAr3] likely proceeds via the heterolytic cleavage of $H_2$ by the FLP species that are formed following the regeneration of the N-phosphinoyl oxygen–borane adducts. These steps are expected to be the rate-determining events because the concentration of $H_2$ (Fig. 4a), the steric bulk of the N-aryl group[17] and the Lewis acidity of the BAr3 moiety (Fig. 3b) influence the reaction rates and/or the temperature required to initiate the reaction between the carbene–borane adducts and $H_2$.

**Theoretical studies.** Density-functional theory (DFT) calculations were carried out at the $\omega$B97X-D/6-311G(d,p), PCM (DCE)// $\omega$B97X-D/6-31G(d,p) for $H_2$ and 6-31G(d) for all other atoms level of theory (Fig. 5a). The relative Gibbs free energies with respect to [**1a** + **B1**] (0.0 kcal·mol$^{-1}$) are shown. During the transformation of **3aB1** (−17.2 kcal·mol$^{-1}$) to **2aB1** (−9.8 kcal·mol$^{-1}$), both of which were experimentally confirmed, the formation of an intermediate **2a'B1** (−7.7 kcal·mol$^{-1}$) was predicted via a C-to-O transfer of **B1** in **3aB1**. This distinctive boron-transfer process takes place via saddle

**a**

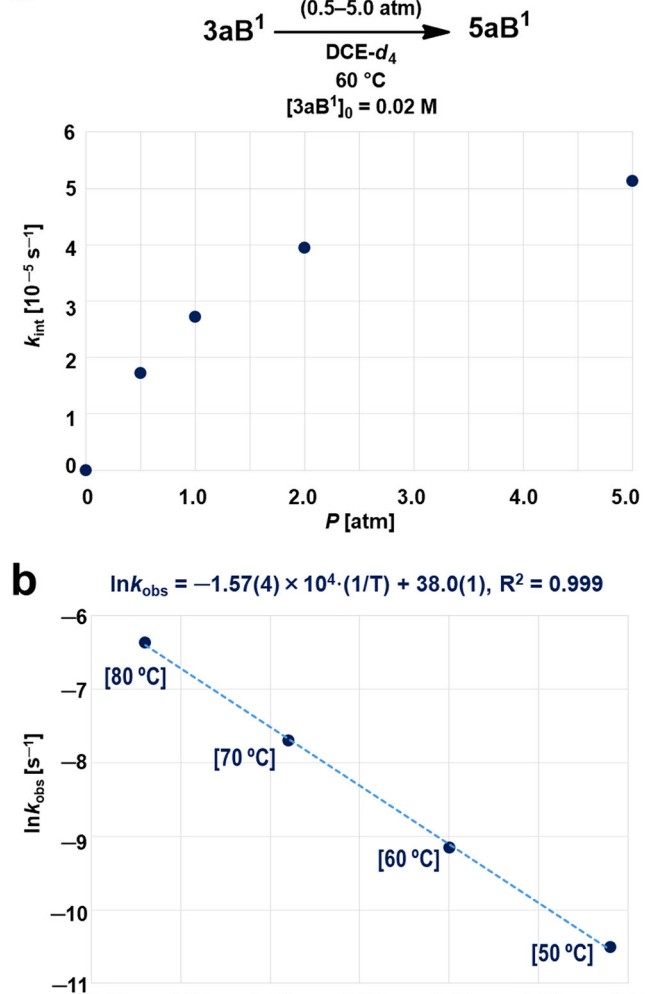

$$3aB^1 \xrightarrow[\substack{DCE\text{-}d_4 \\ 60\,°C \\ [3aB^1]_0 = 0.02\ M}]{\substack{H_2 \\ (0.5\text{--}5.0\ atm)}} 5aB^1$$

**b**

$$\ln k_{obs} = -1.57(4) \times 10^4 \cdot (1/T) + 38.0(1),\ R^2 = 0.999$$

**Fig. 4 Kinetic studies for the reaction between 3aB$^1$ and H$_2$. a** Plot of the H$_2$ pressure, $P$ [atm], as a function of the initial reaction rate constants, $k_{int}$ [10$^{-5}$ s$^{-1}$]. **b** Plot of $1/T$ [10$^{-3}$ K$^{-1}$] as a function of ln$k_{obs}$ [s$^{-1}$]. The $k_{obs}$ values are the pseudo-first order rate constants for the formation of **5aB$^1$** obtained from the reaction of **1a** (2.0 × 10$^{-2}$ M in DCE-$d_4$) and H$_2$ (5 atm).

point **TS1a** (+7.3 kcal·mol$^{-1}$), while the potential energy surface around **TS1a** is very flat (Supplementary Figure 34 for details). The subsequent rotation of the $N$-phosphinoyl moiety via **TS2a** (+7.5 kcal·mol$^{-1}$) affords **2aB$^1$**. Next, the dissociation of the B–O bond occurs to regenerate [**1a** + **B$^1$**]. The optimized molecular structures of **TS1a** and **2a′B$^1$** are shown in Fig. 5c. In **TS1a**, the interatomic distances C1···B and O···B are 4.24 and 3.34 Å, respectively, while **B$^1$** adopts a planar geometry. Thus, **B$^1$** is dissociated from both the carbene carbon and phosphinoyl oxygen atoms in **TS1a**, while the formation of the O–B bond (1.59 Å) is confirmed in **2a′B$^1$**. Based on the quantum theory of atoms in molecule (AIM) method, neither bond paths nor bond critical points were confirmed between the B and C1/O atoms in **TS1a** (Supplementary Figure 37)[33,34]. This AIM analysis demonstrates that several non-covalent interactions, including π–π and H···F interactions, exist between the **1a** and **B$^1$** moieties to stabilize **TS1a**.

Two plausible mechanisms were evaluated for the FLP-mediated cleavage of H$_2$ on the basis that the Lewis-basic center reacts with H$_2$ via cooperation with **B$^1$** (Fig. 5b). One possibility is that the carbene carbon atom works as a Lewis base (path I; the

right path in Fig. 5b)[1–9,35,36], while the other is that the $N$-phosphinoyl oxygen functions as a Lewis base (path II; the left path in Fig. 5b)[37]. In path I, the heterolytic cleavage of H$_2$ takes places via **TS4a** (+11.4 kcal·mol$^{-1}$), which arises from the insertion of H$_2$ into the reaction field around the carbene carbon and boron atoms in **FLP-1aB$^1$**, affording **5aB$^1$** (−34.8 kcal·mol$^{-1}$), a species more thermodynamically stable than **3aB$^1$**. In the optimized structure of **TS4a** (Fig. 5d), the dissociation of the H1–H2 bond (H1···H2 = 0.84 Å) occurs with the partial formation of the H2–C1/H1–B bonds (H2···C1 = 1.83 Å/H1···B = 1.49 Å). Based on these results, the overall path from **3aB$^1$** to **5aB$^1$** via **FLP-1aB$^1$** is substantially exothermic ($\Delta G° = -17.6$ kcal·mol$^{-1}$) and includes an overall activation energy barrier of +28.6 kcal·mol$^{-1}$ required to overcome **TS4a**. In path II, which takes place via **TS5a** (a transition state for the insertion of H$_2$ into the O–P bond) and **TS6a** (a transition state for the cleavage of H$_2$ between the O and P atoms), a higher activation energy barrier of +32.7 kcal·mol$^{-1}$ is predicted to yield intermediate **8aB$^1$**, which contains a P=O–H$^+$ and B–H$^-$ species. It should be noted that the potential energy of the optimized **TS6a** (−3633.288355 hartree) is almost identical to that of the optimized **7aB$^1$** (−3633.288363 hartree), which causes the reversed Gibbs energy levels as shown in Fig. 5b after the Gibbs energy correction and implementation of solvent effect. Therefore, the discussion on the activation energy barrier to overcome **TS6a** from **7aB$^1$** should be not essential. The subsequent transfer of H$^+$ from the $N$-phosphinoyl oxygen atom to the carbene carbon atom furnishes **5aB$^1$**, although the details of this process remain unclear at this point. The molecular structure of **TS6a** shows that the cleavage of the H1–H2 bond (H1···H2 = 0.85 Å) by the $N$-phosphinoyl oxygen and boron atoms occurs in a cooperative fashion (Fig. 5d). Given the experimental and theoretical results reported here, we conclude that path I is the more likely one.

The impact of the $N$-aryl substituents on the activation energy barriers for the regeneration of [**1** + **B$^1$**] was evaluated using calculations on **3bB$^1$**, which contains an $N$-2,4,6-Me$_3$-C$_6$H$_2$ group, as well as **3cB$^1$**, which contains an $N$-3,5-$^t$Bu$_2$-C$_6$H$_3$ group. This afforded $\Delta G^{\ddagger}$ values of +28.3 and +32.8 kcal·mol$^{-1}$ for **3bB$^1$** and **3cB$^1$**, respectively (Fig. 5a). These results are consistent with the experimental observations, i.e., that **3aB$^1$**−**3cB$^1$** did not react in the presence or absence of H$_2$ under ambient conditions under the applied conditions. Furthermore, these results might rationalize the fact that temperature to induce the reaction between these CLAs and H$_2$ increases in the order **3aB$^1$** (60 °C) < **3bB$^1$** (80 °C) < **3cB$^1$** (120 °C)[17].

**Conclusion**. In summary, the reaction mechanism for the revival of frustrated carbene–borane pairs from external-stimuli-responsive classical Lewis adducts (CLAs), comprised of $N$-phosphine-oxide-substituted imidazolylidene (PoxIm) and triarylboranes (BAr$_3$), is reported based on a combination of experimental and theoretical studies. Remarkably, a transfer of the borane moiety from the carbene carbon atom to the $N$-phosphinoyl oxygen atom was identified as a key step in the heterolytic cleavage of H$_2$ by the regenerated FLP species. The optimized transition-state structure for this borane-transfer process was confirmed to include no bonding interactions between the carbene carbon/phosphinoyl oxygen and boron atoms, albeit that it is stabilized by intermolecular non-covalent interactions between the PoxIm and BAr$_3$ moieties. The heterolytic cleavage of H$_2$ takes place via the cooperation of the carbene carbon and the boron atoms, and exhibits a lower overall activation energy barrier than that of the path in which a combination of the $N$-phosphinoyl oxygen and boron atom mediates the H$_2$ cleavage. These results demonstrate the essential role of dynamic

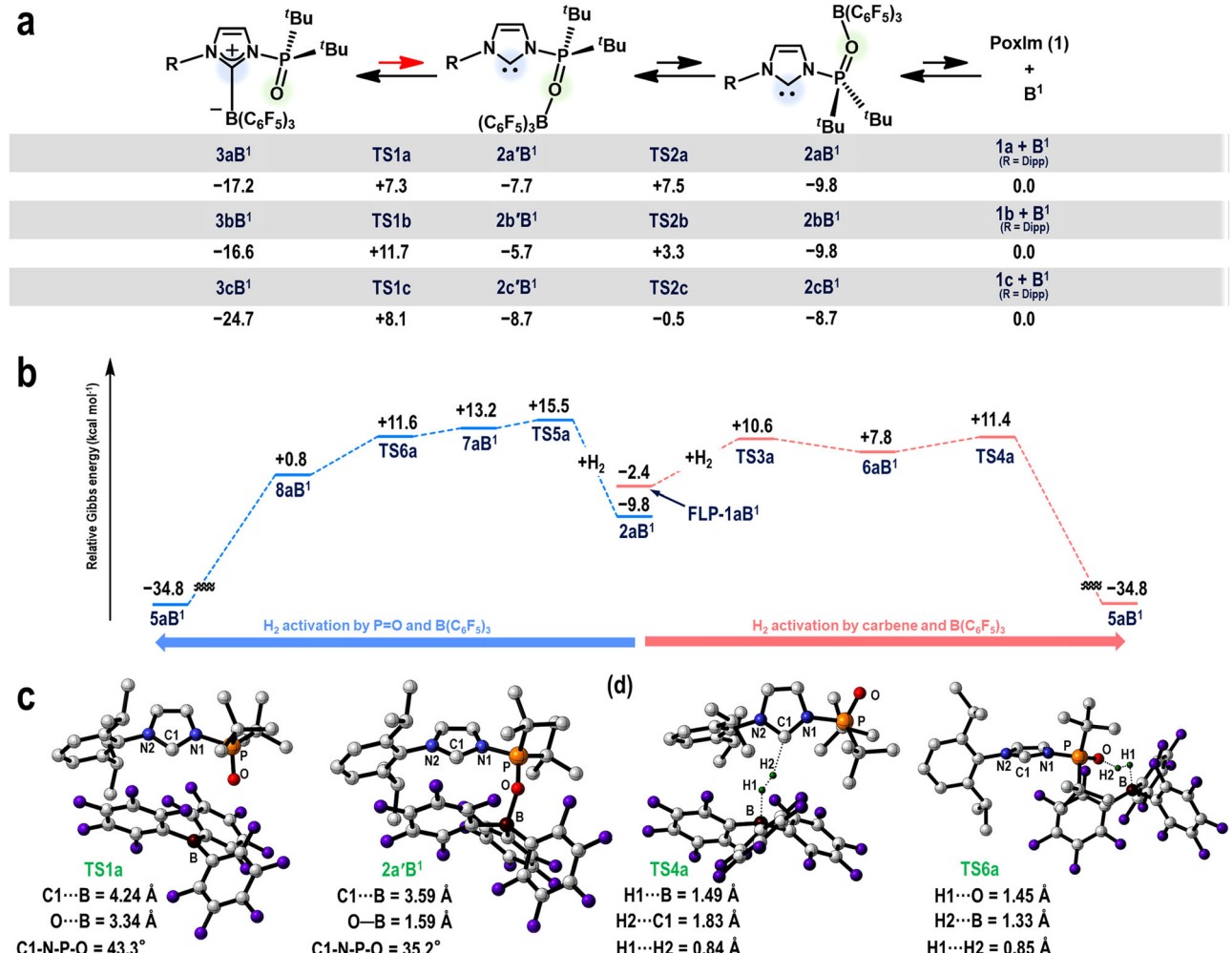

**Fig. 5 Theoretical studies.** The relative Gibbs energies [kcal mol$^{-1}$] are shown with respect to each [**1** + **B$^1$**], calculated at the $\omega$B97X-D/6-311G(d,p), PCM (DCE)//$\omega$B97X-D/6-31G(d,p) (for H$_2$) and 6-31G(d) (for all other atoms) level of theory (298.15 K, 1 atm). **a** Proposed mechanism for the regeneration of [**1** + **B$^1$**] from the carbene–borane complexes **3aB$^1$**–**3cB$^1$**. **b** Proposed mechanism for the heterolytic cleavage of H$_2$, enabled by the phosphinoyl oxygen and B(C$_6$F$_5$)$_3$ moieties (left) or by the carbene and B(C$_6$F$_5$)$_3$ moieties (right). **c** DFT-optimized molecular structures for **TS1a** and **2a′B$^1$**. **d** DFT-optimized molecular structures for **TS4a** and **TS6a**.

conformational isomerization in the regulation of the reactivity of shelf-stable but external-stimuli-responsive Lewis acid-base adducts by multifunctional Lewis bases.

## Methods

**Synthesis of 3aB$^2$.** PoxIm **1a** (154.8 mg, 0.40 mmol) and B($p$-HC$_6$F$_4$)$_3$ (**B$^2$**) (183.4 mg, 0.40 mmol) were mixed in toluene (10 mL) at room temperature to furnish the yellow solution. Stirring this mixture for 4 h resulted into the precipitation of a white solid that was collected via removal of the supernatant solution. The obtained solid was washed with hexane (5 mL) and dried in vacuo to afford **3aB$^2$** as a white solid (230.2 mg, 0.27 mmol, 68%). A single crystal suitable for X-ray diffraction analysis was prepared by recrystallization from CH$_2$Cl$_2$/hexane at −30 °C.

**Synthesis of 5aB$^2$.** A solution of **3aB$^2$** (51.6 mg, 0.06 mmol) in CH$_2$Cl$_2$ (3 mL) was transferred into an autoclave reactor, which was then pressurized with H$_2$ (5 atm). Subsequently, the reaction mixture was stirred at 60 °C for 4 h, before the solvent was removed in vacuo to give **5aB$^2$** as a white solid (51.8 mg, 0.06 mmol, >99%). A single crystal suitable for X-ray diffraction analysis was prepared by recrystallization from THF/hexane at −30 °C.

**Reaction between 1a and B$^2$ giving 2aB$^2$.** A solution of **1a** (7.4 mg, 0.02 mmol) and **B$^2$** (9.3 mg, 0.02 mmol) in CD$_2$Cl$_2$ (0.5 mL) was prepared at −30 °C and then transferred into a J. Young NMR tube. The quantitative formation of **2aB$^2$** was confirmed at −90 °C by $^1$H, $^{13}$C, $^{19}$F, and $^{31}$P NMR analysis (Supplementary

Figs. 5-8). A single crystal suitable for X-ray diffraction analysis was prepared by recrystallization from toluene/hexane at −30 °C.

## Data availability

The datasets generated during and/or analyzed during the current study are available from the corresponding author on reasonable request. Details for the preparation of materials, monitoring the reactions (Supplementary Figs. 5-30), and theoretical analysis/discussion are provided in the Supplementary Information. Compound characterization data including ESI-MS, elementary analysis, and SC-XRD data are also available in the Supplementary Information. NMR spectra for the isolated compounds are provided in the Supplementary Information. Results of the AIM analysis for **TS1a** are provided in Supplementary Fig. 37, as well as Supplementary Data 1. Metrical data for the solid-state structures are available from the Cambridge Crystallographic Data Centre (CCDC) under reference numbers CCDC2072358 (**3aB$^2$**), 2072359 (**5aB$^2$**), 2072360 (**2aB$^2$**), and 2072638 ([**1a–H**][**HO(B$^2$)$_2$**]) (Supplementary Data 2-5). These data can be obtained free of charge from the CCDC via www.ccdc.cam.ac.uk/data_request/cif.

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

## Acknowledgements

This work was supported by Grants-in-Aid for Young Scientists (JSPS KAKENHI grants JP15K17824 and JP18K14219), Grants-in Aid for Scientific Research on Innovative Areas "Stimuli-responsive Chemical Species (JSPS KAKENHI grant JP15H00943)" and "Precisely Designed Catalysts with Customized Scaffolding (JSPS KAKENHI grants JP15H05803 and 15H05805)," and the Environment Research and Technology Development Fund (No. 1RF-2101) of the Environmental Restoration and Conservation Agency of Japan. Y.H. expresses his special thanks to Prof. Dr. T. Sasamori (University of Tsukuba) for valuable suggestions on this work. T.K. expresses his special thanks to a Grant-in-Aid for JSPS Fellows. M.R. and J.H. appreciate support from the Integrated Research Consortium on Chemical Sciences. A part of the computational work was performed at RCCS in Okazaki, Japan. A part of the study was supported by the Cooperative Research Program of Institute for Catalysis, Hokkaido University. S.O. would like to dedicate this work to Prof. Dr. Christian Bruneau.

## Author contributions

This work was conceptualized and directed by Y.H. with the support of J.H. and S.O. M.S. and T.K. performed the experiments, while Y.H., M.S. and T.K. analyzed these data. M.O., M.R. and J.H. performed the computational analysis. The manuscript was prepared by Y.H. supported by discussions with all the other authors.

## Competing interests

The authors declare no competing interests.
