## [Peer Review File · Communications Chemistry]

Reviewers' comments:

Reviewer #1 (Remarks to the Author):

This manuscript by Hoshimoto, Hasegawa, Ogoshi and co-workers re-investigates the transformation of a phosphine-oxide-substituted NHC containing an O-coordinated fluorinated aryl borane to a C-coordinated species. This species reacts further upon heating with transfer of the phosphine oxide substituent to the C-position or reacts with hydrogen to split the H-H molecule. Initial work (ACIE, 2015, Ref 4c) demonstrated the reaction of phosphine-oxide substituted NHCs with B(C₆F₅)₃, including O-coordination, transformation into a C-coordinated species and reaction with H₂. In the manuscript under review, B(p-HC₆F₄)₃ was used as the borane, and the O-coordinated species was more stable, allowing an X-ray structure to be determined. Conversion to the C-coordinated species was observed along with transfer of the OPR₂ group. The key finding with this manuscript is that heating the C-bound species leads to reactivity, termed external-stimuli-responsive here. Heating usually leads to reactivity at some point, so a more unconventional external stimulus would be more exciting. Transfer of the OPR₂ group was seen (reaction with water as an impurity was also observed) and reaction with H₂ if performed under an H₂ atmosphere. Some mechanistic studies into the reaction with H₂ are then reported. The DFT study shows that H₂ activation via the free carbene has a lower barrier than H₂ activation with the phosphine oxide. Overall, the experiments have been carefully performed and the C-bound species / H₂ split compound were thoroughly characterised, with plenty of information in the ESI. The compound derived from water as an impurity has a curious statement: "which afforded a single crystal of [1a-H][HO(B₂)₂] (66.7 mg, 0.05 mmol, 17%)." This is a rather large crystal! Is this correct? The much less stable O-bound species is very well characterised, considering its reactivity. There is a lot of additional data in the ESI as well.

My main concern with this paper is that the compounds are derivatives of published species, with the interest entirely based on the mechanism. Whilst the evidence for this mechanism is relatively strong, is it of sufficient interest to justify the communication? Therefore, a recommendation is hard to make as while the idea of responsive NHC systems is interesting, a new application would make the underlying mechanism more interesting. I would therefore resubmission to a different journal.

Reviewer #2 (Remarks to the Author):

In their manuscript, Ogoshi and co-workers report a joint experimental-computational (DFT) on the mechanism involved in the thermally induced revival of FLPs from N-phosphine oxide-substituted imidazolylidenes and triarylboranes. Although this process was originally reported by some of the authors in 2015 (ACIE 2015,54, 11666) and exploited in different contexts (see works cited in reference 5), the involved mechanism is not fully understood so far. For this reason, the contents of this work, which has been carefully and competently carried out, although rather specific, might attract the attention of the readers of Commun. Chem. Therefore, I support its acceptance. Despite that, the following issues should be addressed in a (major) revision:

(i) Perhaps I missed something, but according to the experimental data, the initially formed 2aB₂ species is converted into 3aB₂ as the reaction progresses. The graphical abstract and the data in

Figure 4a seem to indicate the opposite. So, the first step involves the rotation which is followed by the B-migration. This should be corrected in both the main and the Figures.

(ii) The located TS1a transition state exhibits rather long B...O and, particularly, B...C distances. I am a bit skeptical about the correctness of this saddle point. So, I would suggest to perform an IRC calculation to confirm that this transition state really connects 2aB and 3aB. This IRC plot should be also provided in the supplementary material.

(iii) The identity of TS3a and 6aB (for path I) and TS5a and 7aB (for path II) in the hydrogenation pathways should be also provided. In addition, it is highly surprising that TS6a lies below 7aB (which would indicate a negative activation barrier!!!). This should be checked and if confirmed, a reasonable explanation for that should be given.

(iv) Regarding the computational methodology, for medium-size systems like those considered in this study, at least single-point energy refinements using triple-zeta basis-sets are mandatory.

Responses to Referee 1

Thank you very much for your critical and constructive suggestions. Please kindly confirm the following answers and the corresponding revisions in the revised manuscript.

Q1: Overall, the experiments have been carefully performed and the C-bound species / H₂ split compound were thoroughly characterised, with plenty of information in the ESI. The compound derived from water as an impurity has a curious statement: “which afforded a single crystal of [1a-H][HO(B₂)₂] (66.7 mg, 0.05 mmol, 17%).” This is a rather large crystal! Is this correct?

A1: Thank you for this important note. As mentioned in the corresponding experimental part shown in the Supplementary Materials, we have used commercially available dehydrated toluene, which contains a certain amount of H₂O (~1 ppm), for the synthesis and isolation of [1a-H][HO(B²)₂], while other experiments were conducted using carefully distilled toluene (as mentioned in the Supplementary Materials). We should have emphasized this difference in order to avoid any confusion of the readers, and *the corresponding part is now emphasized in italic style.*

Q2: My main concern with this paper is that the compounds are derivatives of published species, with the interest entirely based on the mechanism. Whilst the evidence for this mechanism is relatively strong, is it of sufficient interest to justify the communication? Therefore, a recommendation is hard to make as while the idea of responsive NHC systems is interesting, a new application would make the underlying mechanism more interesting. I would therefore resubmission to a different journal.

A2: Thank you very much for kind suggestions. We should have mentioned that **this manuscript was submitted as a potential “Article” for *Communications Chemistry*, which would require a relatively comprehensive mechanistic study.** In addition to the unprecedented observation of the boron migration between two different Lewis-basic sites, the novelty of our results, i.e., the stimuli-responsive molecular dynamics that can potentially be recognized in terms of a molecular machine that enables connecting energetically well separated species (Lewis acid-base adducts and their frustrated pairs, can be expected to have considerable impact on the field of organic, organometallic, and main-group chemistry; a notion that you and Referee 2 have already kindly agreed with. Therefore, we think that the revised manuscript is now suitable for publication in *Communications Chemistry*; respectfully, we

would like to ask you to reconsider this point.

Responses to the Referee 2

We would like to thank referee 2 for his/her very positive evaluations as well as critical suggestions on the theoretical experiments. We have carried out the requested single-point energy refinements using triple-zeta basis sets. Please confirm the details as shown below.

Q3: Perhaps I missed something, but according to the experimental data, the initially formed 2aB2 species is converted into 3aB2 as the reaction progresses. The graphical abstract and the data in Figure 4a seem to indicate the opposite. So, the first step involves the rotation which is followed by the B-migration. This should be corrected in both the main and the Figures.

A3: Thank you for your comments. We have shown that the transformation of both **2aB²** to **3aB²** (Fig. 2a) as well as **3aB²** to **2aB²** (Fig. 3) occurred under the applied conditions. The reversibility of these transformations was supported by DFT calculations, and **we thus showed reversible processes in Fig. 5a**. To emphasize this unprecedented boron-migration process (**3aB²→2aB²**), we had attached a graphical abstract; however, **we have removed the corresponding graphical figure in conformity with the editorial policy of *Communications Chemistry***.

Q4: The located TS1a transition state exhibits rather long B···O and, particularly, B···C distances. I am a bit skeptical about the correctness of this saddle point. So, I would suggest to perform an IRC calculation to confirm that this transition state really connects 2aB and 3aB. This IRC plot should be also provided in the supplementary material.

A4: The potential energy surface around **TS1a** was found to be relatively flat (Figure A, also given as Supplementary Fig. 34). Ordinary IRC calculations were thus difficult to apply. To answer this comment, we performed a relaxed potential energy scan calculation that approximately shows the connection from **2a'B¹** to **3aB¹** via **TS1a**:

Figure A. Minimum energy potential pathway from **2a'B¹** to **3aB¹** via **TS1a**, obtained from a relaxed potential energy scan calculation. For the reaction coordinate from **2a'B¹** to **TS1a**, the B-O(P=O) distance was used. For that from **3aB¹** to **TS1a**, the B-C(carbene) distance was used.

We have added these results to the Supplementary Materials.

Q5: The identity of **TS3a** and **6aB** (for path I) and **TS5a** and **7aB** (for path II) in the hydrogenation pathways should be also provided. In addition, it is highly surprising that **TS6a** lies below **7aB** (which would indicate a negative activation barrier!!!). This should be checked and if confirmed, a reasonable explanation for that should be given.

A5: Regarding the identity of **TS3a** and **6aB¹**: Relative potential energies are shown in Figure B(a) (also given as Supplementary Fig. 35a) for the structures along a minimum energy pathway. To the left of **TS3a**, the result of an IRC calculation is shown. To the right of **TS3a**, the relative potential energy is plotted for structures along the steepest decent pathway that was obtained from a structural optimization with very small increments. This optimization terminated at a metastable minimum which is energetically 0.4 kcal/mol higher than the **6aB¹** state. This metastable point is structurally close to that of the **6aB¹** state (Figure B(b) and (c), also given as Supplementary Figs. 35b and 35c). Therefore, under thermal fluctuations in the experimental conditions, the system can be expected to reach the **6aB¹** state.

Figure B. (a) Relative potential energies of structures along a minimum energy pathway via **TS3a**. Calculations were performed with the ω B97XD functional without including solvation effects. Basis sets of 6-31G(d,p) for H₂ and 6-31G(d) for the other atoms were used. For the reaction coordinates from 0 (**TS3a**) in negative direction, an IRC is given. Those from 0 in positive direction represent potential energies of structures along the steepest descent direction obtained from a structural optimization with very small increments. For the **6aB¹** state, a reaction coordinate of 90 is given only for representation purposes. The numbers in parentheses are potential energy values relative to that of **TS3a**. Structures of (b) the metastable minimum state and (c) the **6aB¹** state with selected structural parameters.

Regarding the connection from **TS5a** and the **7aB¹** state: Relative potential energies are shown in Figure C (also given as Supplementary Fig. 36) for the structures along a minimum energy pathway. To the right of **TS5a**, the result of the IRC calculation is given. To the left of **TS5a**, the relative potential energy is plotted for structures along the steepest decent pathway obtained

from a structural optimization with very small increments. This optimization terminated at the **7aB¹** state, which shows the connectivity between the two stationary points.

Figure C. Relative potential energies of structures along a minimum energy pathway via **TS5a**. Calculations were performed with the ω B97XD functional without including solvation effects. Basis sets of 6-31G(d,p) for H₂ and 6-31G(d) for the other atoms were used. For reaction coordinates from 0 (**TS5a**) in positive direction, the result of the IRC calculation is given. Those from 0 in negative direction are potential energies of structures along the steepest descent direction given by structural optimization with very small increments. The numbers in parentheses represent potential energy values relative to that of **TS5a**.

Given the results on the structural optimization, the potential energy of **TS6a** (-3633.288355 hartree) is found to be very close to that of **7aB¹** (-3633.288363 hartree). In general, Gibbs energy correction (Δ Gibbs) is positive, and Δ Gibbs to a transition state is smaller than that of an equilibrium state. This causes the reversed energy level between **TS6a** and **7aB¹** found in Figure 5b. The energy difference after the Gibbs correction and the implementation of the solvent effects was 1.5 kcal/mol, of which 1.4 kcal/mol arises from Δ Gibbs.

We should have clearly mentioned these discussion as you pointed out. Please kindly confirm that we have added these discussions of the results shown in Figures B and C in the revised Supplementary Materials as Supplementary Figs 35 and 36, respectively. In addition, the discussion on the energy difference between **TS6a** and **7aB¹** was added to the main text as follows.

“It should be noted that the potential energy of **TS6a** (-3633.288355 hartree) is almost identical to that of **7aB¹** (-3633.288363 hartree), which causes the reversed Gibbs energy levels as shown in Figure 5b after the Gibbs energy correction and the implementation of the solvent effects.

Therefore, the discussion on the activation energy barrier to overcome **TS6a** from **7aB¹** should be not essential.”

Q6: Regarding the computational methodology, for medium-size systems like those considered in this study, at least single-point energy refinements using triple-zeta basis-sets are mandatory.

A6: As suggested, we should have used triple-zeta basis sets for the discussion of the relative Gibbs free energies. **We have conducted the energy refinements at the ω B97XD/6-311(d,p), PCM (DCE) level of theory, and updated the related discussion.** Please kindly confirm the revised Fig. 5 for details:

Fig. 5 Theoretical studies. The relative Gibbs energies [kcal mol⁻¹] are shown with respect to each [1 + B¹], calculated at the ω B97X-D/6-311G(d,p), PCM (DCE)// ω B97X-D/6-31G(d,p) (for H₂) and 6-31G(d) (for all other atoms) level of theory (298.15 K, 1 atm). **a**) Proposed mechanism for the regeneration of [1 + B¹] from the carbene–borane complexes **3aB¹–3cB¹**. **b**)

Department of Applied Chemistry, Faculty of Engineering, Suita, Osaka 565-0871, Japan

Phone +81-6-6879-7393, Fax +81-6-6879-7394

E-mail hoshimoto@chem.eng.osaka-u.ac.jp, ogoshi@chem.eng.osaka-u.ac.jp

Proposed mechanism for the heterolytic cleavage of H₂, enabled by the phosphinoyl oxygen and B(C₆F₅)₃ moieties (left) or by the carbene and B(C₆F₅)₃ moieties (right). **c**) DFT-optimized molecular structures for **TS1a** and **2a'B¹**. **d**) DFT-optimized molecular structures for **TS4a** and **TS6a**.

REVIEWERS' COMMENTS:

Reviewer #2 (Remarks to the Author):

The authors have addressed the issues listed in my previous report in a satisfactory way. Therefore, I am glad to support the acceptance of this nice work.

Despite that, it is a bit unfortunate that the authors could not perform the IRC for the key transition state TS1a. The relaxed scan shown in the revision is misleading because the reaction coordinate is projected onto two different distances. It would be helpful if this figure could be revised using the same geometrical parameter (or split the Figures into 2 different reaction coordinates).

Anyway, I would suggest to warn the readers (in the main text) that the potential energy surface around TS1a is really flat. In addition, please change the term "activation complex" to "saddle point" or "transition state" in the text.

Once these very minor issues are addressed, the manuscript should be accepted without further modifications.

August 24th, 2021

Responses to Referee 2

Thank you very much for your kind suggestions. Please kindly confirm the following answers and the corresponding revisions in the revised manuscript.

Q1: Despite that, it is a bit unfortunate that the authors could not perform the IRC for the key transition state TS1a. The relaxed scan shown in the revision is misleading because the reaction coordinate is projected onto two different distances. It would be helpful if this figure could be revised using the same geometrical parameter (or split the Figures into 2 different reaction coordinates).

A1: We have revised Supplementary Figure 34 as shown below:

Q2: I would suggest to warn the readers (in the main text) that the potential energy surface around TS1a is really flat. In addition, please change the term "activation complex" to "saddle point" or "transition state" in the text.

A2: Thank you for your kind suggestions. As suggested, we have exchanged the term "transition state" with "saddle point" and added the corresponding comments for potential energy surface as follows in page 7:

"This distinctive boron-transfer process takes place via **saddle point TS1a** (+7.3 kcal·mol⁻¹), while the potential energy surface around **TS1a** is very flat (Supplementary Figure 34 for details)."